# Physical, Organoleptic, and Phytochemical Valuation of the Promising Greek Kiwifruit Genotype Arta Kiwifruit

**Anna Kosta** [1], **Nikoleta-Kleio Denaxa** [1,*], **Athanasios Tsafouros** [1], **Efstathios Ntanos** [1], **Efthalia Stathi** [2], **Eleni Tani** [2] **and Peter Anargyrou Roussos** [1,*]

[1] Laboratory of Pomology, Department of Crop Science, Agricultural University of Athens, 118 55 Athens, Greece; annakosta95@gmail.com (A.K.); thantsaf@hotmail.com (A.T.); stathntan@hotmail.gr (E.N.)

[2] Laboratory of Plant Breeding and Biometry, Department of Crop Science, Agricultural University of Athens, 118 55 Athens, Greece; lilian_stathi@yahoo.gr (E.S.); etani@aua.gr (E.T.)

[*] Correspondence: nkdenaxa@aua.gr (N.-K.D.); roussosp@aua.gr (P.A.R.)

**Abstract:** To evaluate a new kiwifruit genotype named "Arta Kiwifruit", a comparative analysis of fruit physical characteristics and biochemical composition in comparison to the widely cultivated Hayward cultivar took place, both at harvest and after a storage period for two successive years. The findings revealed that "Arta Kiwifruit" holds a significant advantage, as it features a larger fruit size, is approximately 34% heavier than "Hayward", and boasts a distinct shape characterized by a bigger length. Both genotypes exhibited similar dry matter content. No significant difference was observed in protein content, or total phenolic compounds between the two genotypes, while "Hayward" exhibited a significantly higher concentration of sugars and higher sweetness index. "Hayward" demonstrated also increased levels of chlorophyll at harvest, along with higher concentrations of carotenoids. "Hayward" during the first year presented higher antioxidant activity both at harvest and after storage. On the other hand, "Arta Kiwifruit" displayed elevated amino acid concentrations compared to "Hayward", both at harvest and after storage. Phenotypic traits, as well as the genetic analysis using inter-simple sequence repeats (ISSR) markers, further confirmed the distinct genetic profile of "Arta Kiwifruit", highlighting its potential significance for kiwifruit breeding programs and commercial use.

**Keywords:** amino acids; molecular analysis; organic acids; organoleptic characteristics; phenolic compounds; photosynthesis; sugars



## 1. Introduction

Kiwifruit holds a prominent position in the global fruit industry, owing to its distinctive flavor, exceptional nutritional value (high vitamin C content, rich in dietary fiber), and associated health advantages (positive action on cardiovascular diseases, lipid metabolism, inflammatory response, blood pressure problems, overweight, etc., while it is also preferred by vegans) [1]. Kiwifruits are available in a diverse range of cultivars, distinguished by their green, gold, or red flesh as well as their size. Despite the existence of over 60 species within the *Actinidia* genus, only a few have achieved notable economic significance, with the green-fleshed "Hayward" cultivar enjoying widespread cultivation worldwide [2].

The cultivation and production of kiwifruits continue to expand. with leading global producers being China, Italy, New Zealand, and Greece. As many countries increase their kiwifruit production, the kiwifruit industry is emphasizing not only product quality (in terms of dry matter, total soluble solids, and firmness at harvest) but also the development of a market-oriented approach to maintain competitiveness [3]. Furthermore, recognizing that the survival of the industry hinges on innovating new products, significant breeding programs have been launched to formalize the development of new cultivars with desirable characteristics (size, storability, increased firmness during storage and shipment, enhanced

disease resistance, etc.), as well as less vulnerable to climate change (fewer chilling units requirements, etc.) [3]. There is optimism that these endeavors will produce commercially accessible fruits with a variety of sizes, peel characteristics, flesh colors, flavors, nutraceutical properties, and textures to complement the "Hayward" cultivar. Considering the wide geographical distribution and diversity of wild kiwifruit populations, Ferguson [4] anticipates that there will likely be numerous appealing traits that can be integrated into breeding programs.

To evaluate as well as to distinguish different kiwifruit cultivars, both traditional and modern approaches have been applied. Traditional approaches encompass the evaluation of morphological and biochemical traits, including fruit shape, size, color, chemical composition, and flowering time [5]. For instance, Mavromatis [6] distinguished two kiwifruit cultivars based on morphological characteristics such as fruit weight, length, and diameter. Lee [7] and Latocha [8] used biochemical markers such as total phenolic content and antioxidant activity. Nevertheless, it is important to note that these traditional methods have inherent limitations in terms of accuracy and influence exerted by environmental factors.

The use of molecular markers has gained momentum as a reliable technique to distinguish kiwifruit cultivars based on their genetic profiles [6]. This approach has grown in popularity due to its high accuracy, reproducibility, and independence from environmental and agronomic factors. Various molecular markers, including simple sequence repeats (SSR), random amplified polymorphic DNA (RAPD), and inter-simple sequence repeats (ISSR), have been widely employed for this purpose [9–13]. Among these options, ISSRs have garnered particular attention due to their high polymorphism levels, reproducibility, and cost-effectiveness in identifying and classifying kiwifruit cultivars [12,13]. Furthermore, inter-simple sequence repeat (ISSR) markers have been successfully used to ascertain the genetic fidelity among mother plants and their in vitro multiplied kiwifruit regenerates [13].

"Arta Kiwifruit" is a clonal selection originating from the renowned "Hayward" cultivar, found as a random plant within a "Hayward" orchard by the farmer Mrs. Xylogianni Evanthia, in Arta City, Greece. This unique genotype was distinguished and was specifically selected for its exceptional ability to produce large-sized fruits, making it a valuable candidate for commercialization and promotion. Therefore, the primary objective of this study was to evaluate some physical and biochemical traits of the new kiwifruit genotype both at harvest and after storage. Additionally, the genetic difference between the widely cultivated "Hayward" cultivar and the new Greek kiwifruit selection "Arta Kiwifruit", was also examined using molecular markers, to ensure the uniqueness of this new genetic material. Part of the evaluation process was also to compare the bioactive compounds of this genotype with those of the well-known and widely consumed "Hayward" cultivar.

## 2. Materials and Methods

### 2.1. Study Site, Plant Material, and Experimental Design Description

The study was conducted in the city of Arta (latitude 39.158241 and longitude 20.987684), a major kiwifruit cultivation area in Greece, located in the Epirus region of Western Greece, during two successive years (2019 and 2020). The plant material consisted of mature vines of the "Hayward" cultivar and a clonal selection of "Hayward" (referred to in the manuscript as "Arta Kiwifruit"), which was recommended by the farmer Mrs. Xylogianni Evanthia (Arta City, Greece). The plants were propagated from cuttings and grown in an open field. The study was carried out in 1-hectare grower-managed orchard and kiwifruit production was monitored. Both cultivars of kiwifruit vines had a trunk height of 1.8 m and were trained in a T-shape system, with a planting distance of 2.0 m × 4.0 m. Plants were seven years old and all cultivation practices, including irrigation, fertilization, pruning, weeding, and the application of fungicides and pesticides, were consistent across all vines in the orchard, in order to ensure plant health and productivity.

The experiment was arranged as a complete randomized design, with five replications of two vines per cultivar. In total, 20 plants (10 plants per cultivar) were used, i.e., 2 cultivars × 5 replications × 2 plants per replication.

### 2.2. Fruit Sampling and Analysis of Commercial Quality Characteristics

Crop harvest took place in late October when all fruits from the vines of cultivars were harvested at the stage of commercial maturity. Then, at least 20 fruits per replicate (as a representative fruit sample per plot) were randomly sampled from the harvested fruits (avoiding unhealthy, damaged, or misformed fruits), placed into labeled plastic bags, and immediately transferred via a portable freezer to the laboratory to perform further fruit phytochemical analyses.

At the laboratory, the weight of each fruit along with its diameter, and length were determined with an electronic balance (Kern 470, Kern and Sohn, GmbH, Stuttgart, Germany) and a digital caliper (Starrett, 727 Series, Athol, MA, USA). Firmness was measured at the two opposite sides of each fruit with a penetrometer (Turoni 53205 fruit pressure tester) (T.R. Turoni srl, Forlì, Italy) with a conical tip, after peeling a small part of the fruit skin using a sharp knife. The flesh color of each fruit was assessed at two opposing points on the equatorial region using a Minolta CR 300 reflectance Chroma Meter (Minolta, Osaka, Japan) after exposing fruit flesh, by peeling off a portion of skin. The Chroma Meter provided CIE L*, a*, and b* values, which were used to calculate hue angle degree ($h° = \arctan [b\ a^{-1}]$), where 0° = red–purple; 90° = yellow; 180° = bluish-green and 270° = blue as well as Chroma color index ($C* = [a^2 + b^2]^{1/2}$), indicative of the intensity or color saturation.

The dry matter percentage of eight fruits per replicate was determined after drying in an oven at 70 °C to constant weight an approximately 5 mm thick fruit portion derived from the equatorial portion of the fruit. The rest of the fruits were peeled and homogenized in a household homogenizer. The pulp was then placed in 50 mL tubes and stored in a freezer at −25 °C till further analyses.

The kiwifruit juice was analyzed to determine the total soluble solids (TSS) by the use of a digital refractometer (Hanna HI96801, Hanna Instruments, Smithfield, RI, USA), total titratable acidity (TA) (juice diluted 1:20 with HPLC grade water and titrated with NaOH 0.05 N, till pH 8.2), and pH, according to the method described by Denaxa et al. [14]. TSS was expressed as °Brix, while TA was expressed as a percentage (*w*/*v*) of citric acid present in the juice.

The rest of the harvested fruits were stored in cold rooms under 0.5 °C and 90–95% humidity for 3 months. At the end of the storage period, at least 20 fruits per replicate were randomly collected and the physiological properties were measured as mentioned previously. Some of these fruits were peeled and homogenized as previously described to study the post-storage fruit quality characteristics.

### 2.3. Soluble Sugars and Starch Determination

The frozen pulp (2 g) underwent two extractions using 4 mL of HPLC-grade water in a microwave, following the method of Roussos et al. [15]. The separation of sucrose, glucose, fructose, and inositol was performed using a Shimadzu Nexera X2 HPLC system equipped with an LC 30AD pump and a refractive index detector (Hewlett Packard HP1047A) (Agilent, Santa Clara, CA, USA). An Adamas Amino 5 μm column (250 mm × 4.6 mm) (Sepachrom, Milan, Italy), maintained at 35 °C, was used for the analysis, with acetonitrile: water (80:20) serving as the mobile phase at a flow rate of 1.0 mL min$^{-1}$. The total sugar concentration was determined by summing the concentrations of the individual sugars detected through HPLC. Each sample was analyzed twice, and the final concentrations were expressed as mg per g of fresh weight (FW).

The sweetness index (SI) of the fruit was calculated as:

SI = 1.00 × (glucose concentration) + 1.35 × (sucrose concentration) + 2.3 × (fructose concentration) + 0.685 × (inositol concentration) [15].

The obtained pellet from the sugar determination was washed twice with 80% $v/v$ ethanol in water, followed by an additional wash with pure ethanol, and allowed to dry. Subsequently, the starch concentration in the pellet was estimated through enzymatic hydrolysis, following the method described by Denaxa et al. [16]. The absorbance of the resulting pink solution was measured at 510 nm against a blank, and the starch content was quantified using a five-point calibration curve created using original corn starch treated as mentioned earlier. Each sample underwent duplicate analyses and was expressed as mg per g of fresh weight (FW).

### 2.4. Organic Acids Determination

The analysis of organic acids was performed using a Shimadzu Nexera X2 system, equipped with a diode array detector (DAD) (SPD-M20A, Shimadzu, Kyoto, Japan) at a wavelength of 210 nm. The frozen pulp was extracted twice using a 3% $w/v$ metaphosphoric acid solution in water, following the method described in Roussos et al. [15]. The mobile phase consisted of 0.02% $v/v$ formic acid in water, and the analysis was performed isocratically at a flow rate of 1.5 mL min$^{-1}$ at 27 °C using a Kinetex C18 EVO column (250 mm × 4.6 mm, 5 μ) (Phenomenex, Torrance, CA, USA). Citric acid, malic acid, and ascorbic acid were identified, while the total organic acids concentration was calculated by summing the concentrations of the individual acids and expressed as mg per g of fresh weight.

### 2.5. Phenolic Compounds and Antioxidant Capacity Determination

Phenolic compounds were extracted twice from approximately 1 g of frozen pulp with 3 mL of methanol in a water bath at 38 °C following the method of Roussos et al. [15]. The total phenolic content was analyzed by the Folin–Ciocalteu method, *o*-diphenol content by the sodium molybdate method, the total flavonoids by the sodium nitrite–aluminum chloride method, and the total flavanols by the dimethylamino cinnamaldehyde method. The methanolic supernatants were used to determine the total phenols, total o-diphenols, total flavanols, and total flavonoids, as described in Denaxa et al. [14]. Total phenols were expressed as mg equivalent of gallic acid (GAE), total *o*-diphenols as μg equivalent of caffeic acid (CAE), total flavanols as μg equivalent of catechin (CtE), and total flavonoids as μg equivalent of CAE per g of fresh weight, respectively.

The antioxidant capacity was assessed using the ABTS (2,2-azino-bis (3-ethylbenzothiazoline-6-sulfonic acid), DPPH (2,2-diphenyl-1-picrylhydrazyl), and FRAP (ferric reducing antioxidant power) assays, following the procedure outlined in Denaxa et al. [14]. These assays were conducted on the same methanolic fraction used for phenolic compound analysis. The antioxidant capacity was expressed in micromoles (μmol) of Trolox equivalents (TE) per g of fresh weight.

### 2.6. Chlorophylls and Carotenoids Determination

Total chlorophylls and carotenoids were determined by extracting approximately 2 g of frozen pulp twice with 5 mL of 80% $v/v$ acetone in water. Then, the two supernatants were combined, and the absorbance of the solution was measured at 663 nm and 646 nm. Chlorophyll concentration was measured according to the formulas described by Lichtenthaler [17] (Chla = 12.25 × Abs663 − 2.79 × Abs645, Chlb = 21.50 × Abs645 − 5.10 × Abs663, Chla + b = 7.15 × Abs663 + 18.71 × Abs645).

The carotenoids were determined at a wavelength of 470 nm and their concentration was determined using the formula also described by Lichtenthaler [17] [Carotenoids = (1000 × Abs 470 − 3.27 × Chla − 104 × Chlb)/229]. Both chlorophylls and carotenoids were expressed as μg per g fresh weight.

### 2.7. Amino Acids Determination

For the amino acid analysis, approximately 0.1 g of kiwifruit pulp was extracted twice with 2 mL 0.1 N HCl under continuous stirring for 5 min in an orbital shaker at room

temperature. The extract was centrifuged at $4000 \times g$ for 6 min, the two supernatants were combined. The derivatization of the amino acids was based on the method described by Shi et al. [18] using phenyl isothiocyanate (PITC) as a derivatizing agent. Briefly, 100 μL of 0.1 M PITC in acetonitrile and 100 μL of 1.0 M trimethylamine in acetonitrile were added to 200 μL of extract and mixed thoroughly. The reaction lasted for 1 h and afterwards 400 μL of hexane was added and vortexed. After standing for 10 min, the lower-layer solution was pipetted out and filtered through a 0.45 μm nylon syringe filter. The amino acids were analyzed by HPLC Shimadzu Nexera X2 system, equipped with a diode array detector at 254 nm (DAD) (SPD-M20A, Shimadzu, Kyoto, Japan), using a Luna C18 column (250 mm × 4.6 mm) (Phenomenex, Torrance, CA, USA) running at room temperature at a flow rate of 0.9 mL min$^{-1}$. The mobile phase consisted of A: sodium acetate buffer pH 6.5 in 10% acetonitrile in water and B: acetonitrile–water at 4:1 ($v/v$) ratio. The gradient was set as follows: at 0 min the mobile phase B at 3%, at 7 min B at 9% standing for 10 min, at 25 min B at 20% standing for 10 min, at 60 min B at 38%, at 80 min B at 79,5%, at 85 min B at 100% standing for 10 min, and at 100 min B at 3%.

The following amino acids were identified: alanine, arginine, asparagine, aspartic acid, γ-aminobutyric acid (GABA), glutamic acid, glycine, isoleucine, leucine, methionine, phenylalanine, proline, along with glutamine, tryptophan, threonine, tyrosine, and valine. Quantification was performed using an internal standard (norleucine) and a five-point calibration curve of the amino acids.

### 2.8. Proteins Determination

Two grams of frozen pulp were extracted using an Ultra-Turrax homogenizer with 10 mL of cold 0.1 M phosphate buffer solution pH 7.5, which contained 1 mM ascorbic acid, 2 mM EDTA, and 5% insoluble PVPP. Subsequently, the samples were centrifuged at 4000 rpm for 10 min at 4 °C, and the proteins were determined according to the method described by Bradford [19]. In summary, 0.2 mL of the supernatant was mixed with 5 mL of Coomassie Brilliant Blue G 250 dye (0.01% $w/v$ in 8.5% $w/v$ orthophosphoric acid in water), followed by the addition of 0.8 mL of 0.1 M phosphate buffer pH 7.2. After stirring, the absorbance was measured at 595 nm after 3 min, and the protein concentration in the sample was expressed as mg of bovine serum albumin per g of fresh tissue weight.

### 2.9. Statistical Analysis

The data analysis was performed using the statistical software Statgraphics Centurion XV (Statgraphics Technologies, Inc., The Plains, VA, USA) and JMP13 (SAS Institute Inc., Cary, NC, USA). Significant differences were determined according to Student's *t*-test at $\alpha = 0.05$. Principal component analysis (PCA) after varimax rotation was also performed to describe the effects of cultivars on fruit quality attributes, by a reduced number of factors, separately at harvest and after the storage period per experimentation year.

### 2.10. Phenotypic and Molecular Characterization of Kiwifruit Genotypes

The phenotypic characteristics of the Arta Kiwifruit genotype were recorded in both years, as dictated in the key morphological traits for kiwifruit according to the International Union for the Protection of New Varieties of Plants (UPOV) (https://www.upov.int/edocs/tgdocs/en/tg098.pdf, assessed on 13 October 2023). Ten vines of the "Arta Kiwifruit" and another ten vines of the "Hayward" cultivar grown under the same pedoclimatic conditions were used for the morphological traits' evaluation. The phenotypic description of the vines was based on the characteristics of the plant, shoots, leaves, flowering, flowers, and fruit. Each trait was phenotypically characterized and/or measured in spring (buds, young shoot growth, flowering, and flower characteristics), summer (mature leaves and whole vine), and harvest (fruit).

Healthy, undamaged leaves from characteristic plants of "Hayward" and "Arta Kiwifruit" were sampled during mid-June, during the second year of the study, frozen instantly in liquid nitrogen, and transferred to the laboratory for genetic analysis. The leaf

samples were ground into a powder using a mortar, pestle, and liquid nitrogen. DNA was extracted from 100 mg of leaf tissue using the NucleoSpin Plant II Kit (Macherey-Nagel, Düren, Germany). The extracted DNA was checked for quality and quantity using a NanoDrop microvolume spectrophotometer.

For genome screening, 8 ISSR primer sets were initially used: UBC 807, 810, 811, 841, 844, 861, 864, and 880 (Table 1). The PCR reaction solution was the same for all the primer sets and contained 1X KapaTaq buffer, 2.5 µM MgCl$_2$, 0.3 µM dNTPs, 0.3 µM primer, and 1 unit Taq polymerase. PCR parameters to amplify the genomic DNA were the following: 4 min at 95 °C; followed by 37 cycles of 30 s at 94 °C, 1 min at annealing temperature depending on the primer, and 2 min at 72 °C; followed by 7 min at 72 °C. The annealing temperatures for each primer are presented in Table 2. PCR products were separated on a 2% agarose gel. The differences between the two genotypes were visually highlighted due to the small number of genotypes examined.

**Table 1.** Inter-simple sequence repeats (ISSR) sequences.

| ISSR | Sequence |
| --- | --- |
| UBC-807 | 5′-AGAGAGAGAGAGAGAGAGT-3′ |
| UBC-810 | 5′-GAGAGAGAGAGAGAGAGAT-3′ |
| UBC-811 | 5′-GAGAGAGAGAGAGAGAGAC-3′ |
| UBC-841 | 5′-GAGAGAGAGAGAGAGAGAYC-3′ |
| UBC-844 | 5′- CTCTCTCTCTCTCTCTCTRC-3′ |
| UBC-861 | 5′-ACCACCACCACCACCACC-3′ |
| UBC-864 | 5′-ATGATGATGATGATGATG-3′ |
| UBC-880 | 5′-GGAGAGGAGAGGAGA-3′ |

**Table 2.** Annealing temperatures (T °C) of the selected primers.

| UBC Primer | 807 | 810 | 811 | 841 | 844 | 861 | 864 | 880 |
| --- | --- | --- | --- | --- | --- | --- | --- | --- |
| T (°C) | 50 | 50 | 50 | 50 | 53.5 | 50 | 48 | 45 |

## 3. Results and Discussion

The biochemical composition of kiwifruits at harvest reflects the accumulation of primary metabolites, such as sugars and organic acids, as well as secondary metabolites, including phenolic compounds and other antioxidants. These compounds play a significant role in determining the taste, texture, nutritional value, and potential health benefits of the fruit. During storage, various biochemical changes occur, which can affect the overall quality and consumer acceptance of the fruit. Thus, monitoring and comparing the changes in these compounds at harvest and during storage is crucial for assessing the biochemical dynamics of the new kiwifruit genotype.

Productivity also plays an important role in the selection of a new genotype, since this is the first desirable trait the farmer is looking for. The mean production per vine during the first year differed between the two genotypes with "Hayward" presenting a significantly higher yield than the new genotype (Table 3). During the second year, there were not any significant differences between the genotypes, with the yield being approximately 22.5 Kg per vine for both genotypes.

The fruit physical parameters of the "Hayward" and "Arta Kiwifruit" genotypes are summarized in Table 3. Notably, the mean fruit weight of "Arta Kiwifruit" was significantly higher compared to "Hayward" during both years of experimentation both at harvest and after the storage period. Additionally, fruits of "Arta Kiwifruit" exhibited larger lengths compared to "Hayward" during both years. Sotiropoulos et al. [20] also described a genotype of higher fruit weight than "Hayward", which due to this characteristic was considered commercially interesting. Thus, what truly distinguishes "Arta Kiwifruit" is its large fruit size, providing substantial marketable advantages in terms of fruit appearance. These desirable characteristics played a pivotal role in the selection of "Arta Kiwifruit". It is worth

noting that the final fruit size of "Hayward" can be increased through various horticultural practices, including the application of plant growth regulators such as thidiazuron, forchlorfenuron, gibberellic acid, or dichlorophenoxyacetic acid [21]. "Arta Kiwifruit" produces large fruits without the need for chemical treatments, which is a significant advantage, as both the production costs remain low, and the overall kiwifruit cultivation becomes more sustainable (less use of pesticides and plant growth regulators), while at the same time extra fruit quality is achieved (FAO Codex Alimentarius, Standard for Kiwifruit, CXS 338-2020, adopted in 2020, https://www.fao.org/fao-who-codexalimentarius/committees/committee/related-standards/jp/ accessed on 16 November 2023).

**Table 3.** Effect of the genotype on physical parameters of fruit quality evaluated at harvest and after storage.

| Parameters | Harvest | | After Storage | |
|---|---|---|---|---|
| | Hayward | Arta Kiwifruit | Hayward | Arta Kiwifruit |
| | 1st year | | | |
| Yield (Kg/vine) | 17.9 ± 1.8 a | 15.6 ± 1.2 b | | |
| Weight (g) | 106.2 ± 5.6 b A | 146.7 ± 7.8 a A | 102.2 ± 8.8 b A | 132.5 ± 7.3 a B |
| Diameter minor (mm) | 51.3 ± 7.5 a A | 58.6 ± 1.6 a A | 50.8 ± 10.8 a B | 55.1 ± 3.5 a B |
| Diameter wide (mm) | 62.9 ± 6.6 a A | 63.5 ± 1.8 a A | 60.1 ± 2.0 b A | 60.5 ± 2.5 a A |
| Length (mm) | 82.3 ± 8.5 b A | 92.6 ± 1.5 a A | 75.4 ± 5.8 a A | 92.1 ± 1.6 a A |
| Firmness (N) | 31.9 ± 3.4 b A | 38.9 ± 3.0 a A | 5.6 ± 0.3 a B | 4.8 ± 0.5 a B |
| % Dry matter | 16.1 ± 1.5 a A | 15.3 ± 0.6 a A | 16.6 ± 0.5 a A | 15.2 ± 0.5 a A |
| | 2nd year | | | |
| Yield (Kg/vine) | 22.5 ± 1.6 a | 22.2 ± 1.4 a | | |
| Weight (g) | 99.2 ± 1.4 b A | 130.1 ± 5.2 a A | 98.4 ± 6.8 b A | 119.0 ± 4.0 a B |
| Diameter minor (mm) | 53.5 ± 0.5 a A | 56.2 ± 1.9 a A | 51.1 ± 3.5 b B | 53.8 ± 2.3 a A |
| Diameter wide (mm) | 56.2 ± 0.7 a A | 62.8 ± 1.5 a A | 52.8 ± 0.8 b A | 59.6 ± 1.9 a B |
| Length (mm) | 73.4 ± 1.6 b A | 84.9 ± 3.0 a A | 69.6 ± 5.3 b A | 74.2 ± 1.5 a B |
| Fruit firmness (N) | 25.3 ± 5.8 a A | 21.3 ± 4.8 a A | 5.5 ± 0.3 a B | 5.4 ± 0.3 a B |
| % Dry matter | 17.7 ± 1.2 a A | 17.0 ± 0.6 a A | 16.9 ± 0.6 a A | 17.0 ± 0.3 a A |

Means ± standard deviation in rows with different lowercase letters within the same measurement moment (harvest or after storage) indicate significant differences between genotypes according to Student's *t*-test at $\alpha = 0.05$. Different capital letters indicate significant differences between harvest and after storage for the same genotype according to Student's *t*-test at $\alpha = 0.05$.

In addition to fruit size, factors such as firmness, soluble sugars (TSS), organic acids, and dry matter accumulation play crucial roles in determining fruit quality and the ability to be stored effectively [22]. In the present study, the fruit dry matter did not differ between genotypes and was beyond the lower allowable limit for harvest (15%) (FAO Codex Alimentarius, Standard for Kiwifruit, CXS 338-2020, adopted in 2020, https://www.fao.org/fao-who-codexalimentarius/committees/committee/related-standards/jp/ accessed on 16 November 2023) (Table 3). Similarly, the fruit firmness at harvest ranged from 21.3 N to 38.9 N, with "Arta Kiwifruit" exhibiting higher firmness during the first year. After storage, no significant difference was observed between genotypes. Generally, the commercially approved fruit firmness can vary depending on the specific cultivar and market preferences. Previous studies have indicated that fruit ripeness, as assessed through firmness measurements, can significantly influence consumer preferences [23]. Following the findings of Stec et al. [23], Jaeger et al. [3] concluded that commercial *Actinidia* fruits with firmness ranging from 0.6 to 0.8 Kg (approximately 5.9 N to 7.8 N) were significantly more acceptable to consumers. Furthermore, the life storage of kiwifruit is affected by the extent to which the flesh softens [24] and this was obvious in the present study, as both genotypes presented a significant reduction in fruit firmness during both years. According to Krupa [25], kiwifruits are considered ripe and ready to be consumed when the firmness of the flesh measures below 9.81 N.

No significant difference was observed between genotypes in terms of pH, TA, and TSS: TA ratio, as shown in Table 4. Both "Hayward" and "Arta Kiwifruit" fruits were harvested at the commercial maturity stage, which is determined by reaching the minimum marketing value of 6.2 °Brix TSS content [26]. The TSS content ranged from 6.8 to 7.9 °Brix, with "Hayward" fruits having a statistically significantly higher amount during the first year. After storage, TSS content was doubled, ranging on average from 12.6 to 13.8 °Brix, which is generally considered acceptable by consumers [27,28], with "Hayward" fruits of the second year presenting higher TSS than those of "Arta Kiwifruit". Furthermore, both "Hayward" and "Arta Kiwifruit" are green-fleshed kiwifruits, sharing a similar hue angle and they do not exhibit significant differences in terms of flesh color parameters (Table 4). After the storage period though, significant reductions of all color parameters were determined, in both genotypes, indicative of either chlorophyll breakdown and/or just simply pigment concentration due to water loss, as was evident by the slight fruit weight loss.

**Table 4.** Effect of the genotype on commercial chemical parameters of fruit quality and flesh color attributes evaluated at harvest and after storage.

| Parameters | Harvest | | After Storage | |
|---|---|---|---|---|
| | Hayward | Arta Kiwifruit | Hayward | Arta Kiwifruit |
| | 1st year | | | |
| pH | 3.6 ± 0.1 a A | 3.5 ± 0.2 a A | 3.2 ± 0.1 b B | 3.4 ± 0.1 a A |
| TSS | 7.3 ± 0.6 a B | 6.8 ± 0.7 b B | 13.8 ± 0.8 a A | 13.3 ± 1.1 a A |
| TA | 1.9 ± 0.0 a A | 2.7 ± 0.0 a B | 1.3 ± 0.02 a B | 1.3 ± 0.0 a A |
| TSS:TA | 3.9 ± 1.6 a B | 2.3 ± 1.9 a B | 10.7 ± 3.4 a A | 10.4 ± 3.5 a A |
| *L* * | 66.5 ± 1.9 a A | 65.9 ± 0.9 a A | 56.6 ± 1.7 a B | 57.7 ± 3.0 a B |
| Chroma | 36.5 ± 0.9 a A | 36.6 ± 0.7 a A | 28.9 ± 1.9 a B | 27.8 ± 2.4 a B |
| Hue | 118.6 ± 0.3 a A | 117.0 ± 0.7 b A | 114.9 ± 0.6 a B | 114.6 ± 0.8 a B |
| | 2nd year | | | |
| pH | 3.3 ± 0.2 a A | 3.3 ± 0.0 a A | 3.1 ± 0.04 a B | 3.1 ± 0.1 a A |
| TSS | 7.9 ± 0.1 a B | 7.9 ± 0.1 a B | 13.6 ± 0.1 a A | 12.6 ± 0.1 b A |
| TA | 1.6 ± 0.0 a A | 1.6 ± 1.1 a A | 1.2 ± 0.2 a B | 1.7 ± 0.1 a A |
| TSS:TA | 4.9 ± 0.3 a B | 4.3 ± 0.9 a B | 11.4 ± 0.4 a A | 7.6 ± 0.5 a A |
| *L* * | 61.2 ± 1.1 a A | 63.4 ± 3.0 a A | 58.6 ± 3.9 a A | 54.8 ± 2.2 a B |
| Chroma | 33.1 ± 0.6 a A | 31.6 ± 2.2 a A | 24.5 ± 1.8 a B | 24.1 ± 3.3 a B |
| Hue | 119.0 ± 0.2 a A | 118.6 ± 0.6 a A | 116.8 ± 1.7 a B | 116.0 ± 1.6 a B |

Means ± standard deviation in rows with different lowercase letters within the same measurement moment (harvest or after storage) indicate significant differences between genotypes according to Student's *t*-test at $\alpha = 0.05$. Different capital letters indicate significant differences between harvest and after storage for the same genotype according to Student's *t*-test at $\alpha = 0.05$.

According to Nishiyama et al. [29], the *Actinidia* genus exhibits significant variation in fruit total chlorophyll content, ranging from trace amounts to $4.4 \pm 0.8$ mg 100 $g^{-1}$ FW in a hybrid variety of *A. arguta × deliciosa*. In this study, the total chlorophyll content ranged from 14.7 $\mu g\ g^{-1}$ FW in "Hayward" to 11.7 $\mu g\ g^{-1}$ FW in "Arta Kiwifruit" (Table 5). Similar to previous research by Drummond [30], kiwifruits contain both chlorophylls a and b, with varying levels and ratios between different cultivars. In the current study, the "Hayward" fruits had a significantly higher total Chla content at harvest (9.1 and 8.5 $\mu g\ g^{-1}$ FW during the first and second year, respectively), compared to "Arta Kiwifruit" (7.3 and 7.5 $\mu g\ g^{-1}$ FW, respectively). No significant differences were observed in Chlb content for both genotypes, as well as in the total Chls content at harvest. These findings align with previous studies conducted by Nishiyama et al. [29], Drummond [30], and Pilkington et al. [31].

**Table 5.** Effect of the genotype on kiwifruit's chlorophyll and carotenoid concentrations ($\mu g \, g^{-1}$ FW) at harvest and after storage.

| Parameters | Harvest | | After Storage | |
|---|---|---|---|---|
| | Hayward | Arta Kiwifruit | Hayward | Arta Kiwifruit |
| 1st year | | | | |
| Chla | 9.1 ± 0.1 a A | 7.5 ± 0.1 b A | 6.4 ± 0.1 a B | 6.6 ± 0.1 a B |
| Chlb | 5.7 ± 0.2 a A | 4.3 ± 0.2 a A | 4.1 ± 0.0 a B | 4.3 ± 0.1 a A |
| Chla + b | 14.7 ± 0.3 a A | 11.8 ± 0.3 a A | 10.5 ± 0.1 a B | 10.9 ± 0.3 a A |
| Chla/b | 1.6 ± 0.0 a A | 1.7 ± 0.0 a A | 1.6 ± 0.0 a A | 1.5 ± 0.0 a A |
| Carotenoids | 2.2 ± 0.0 a A | 1.8 ± 0.0 b A | 1.8 ± 0.0 a B | 1.7 ± 0.0 a A |
| 2nd year | | | | |
| Chla | 8.5 ± 0.0 a A | 7.3 ± 0.1 b A | 6.7 ± 0.1 a B | 6.5 ± 0.1 a A |
| Chlb | 5.5 ± 0.0 a A | 4.4 ± 0.1 a A | 3.8 ± 0.1 a B | 3.9 ± 0.0 a A |
| Chla + b | 14.0 ± 0.1 a A | 11.7 ± 0.1 a A | 10.5 ± 0.2 a B | 10.4 ± 0.1 a A |
| Chla/b | 1.5 ± 0.1 a A | 1.7 ± 0.1 a A | 1.8 ± 0.1 a A | 1.7 ± 0.1 a A |
| Carotenoids | 1.9 ± 0.0 a A | 1.8 ± 0.0 b A | 1.7 ± 0.0 a A | 1.6 ± 0.0 a A |

Means ± standard deviation in rows with different lowercase letters within the same measurement moment (harvest or after storage) indicate significant differences between genotypes according to Student's *t*-test at $\alpha = 0.05$. Different capital letters indicate significant differences between harvest and after storage for the same genotype according to Student's *t*-test at $\alpha = 0.05$.

Generally, immature fruits commonly contain chlorophylls, but their levels decline rapidly as the fruit matures and ripens. Consequently, very few fruits retain their green color when ripe [30,31]. However, mature green kiwifruits of the *Actinidia deliciosa* species are an exception, as they are characterized by high concentrations of chlorophyll in the fruit flesh. In this study, the total chlorophyll content, as well as Chla and Chlb levels, decreased during the maturation process in storage conditions in "Hayward" fruits, in accordance with Robertson [32]. Based on the present trial the chlorophyll content of the fruit after the storage period was approximately 10 $\mu g \, g^{-1}$ fruit flesh, which is consistent with the findings of Pilkington et al. [31], indicating that both green-fleshed kiwifruit genotypes have one of the highest concentrations of chlorophyll, second only to avocados [33].

The ratio of Chla to Chlb fluctuated between 1.5 and 1.7 at harvest, and between 1.5 and 1.8 after storage (Table 5). "Hayward" exhibited a significantly higher content of total carotenoids compared to "Arta Kiwifruit" at harvest, while the concentration of carotenoids was similar in both genotypes after the storage period (Table 5).

The level of soluble sugars found in kiwifruit is an important indicator of ripeness. As the fruit ripens, the concentration of sugars increases significantly, while the starch content decreases. This transformation is accompanied by changes in fruit texture as well. Once the fruit reaches a state of ripeness suitable for consumption, the sugars contribute to the sweet taste. However, this sweetness is balanced by the acid composition of the fruit, ensuring a harmonious flavor profile [34,35].

The main sugars found in *Actinidia* fruits are glucose, fructose, and sucrose [30]. Table 6 displays the sugar composition within the "Hayward" and "Arta Kiwifruit" genotypes, revealing nearly equal levels of glucose at harvest, while "Hayward" presented higher levels of fructose (both years) and total sugars (during the second year only), which increased significantly (in both genotypes) after the storage period. Similar findings have been reported by Perez et al. [35] and Sanz et al. [36]. Notably, "Hayward" fruits exhibited significantly higher concentrations of fructose, sucrose, glucose, and total sugars as well as sweetness index after the storage period, during both years of the study. No significant differences were observed for starch though. Nishiyama et al. [34] noted that the levels of total sugars and their ratios vary not only with maturity but also across different kiwifruit cultivars, justifying the present results.

**Table 6.** Effect of the genotype on kiwifruit's carbohydrates (sugars are expressed as mg g$^{-1}$ FW and starch as mg g$^{-1}$ FW) and organic acid concentration (mg g$^{-1}$ FW) at harvest and after storage.

| Parameters | Harvest | | After Storage | |
|---|---|---|---|---|
| | Hayward | Arta Kiwifruit | Hayward | Arta Kiwifruit |
| | 1st year | | | |
| Fructose | 25.5 ± 3.4 a B | 17.2 ± 3.8 b B | 60.5 ± 0.8 a A | 47.6 ± 1.1 b A |
| Glucose | 21.5 ± 4.7 a B | 21.8 ± 2.9 a B | 59.6 ± 0.9 a A | 42.1 ± 0.8 b A |
| Sucrose | 6.5 ± 0.3 a A | 3.9 ± 0.6 a A | 5.4 ± 0.1 a B | 3.2 ± 0.3 b A |
| Inositol | 2.2 ± 0.5 a A | 1.2 ± 0.2 b A | 1.2 ± 0.04 a A | 0.8 ± 0.1 b B |
| Total sugars | 55.7 ± 8.0 a B | 44.1 ± 3.6 a B | 126.7 ± 1.8 a A | 93.7 ± 1.8 b A |
| Sweetness index | 9.04 ± 0.9 a B | 6.7 ± 1.5 b B | 20.7 ± 0.7 a A | 15.6 ± 0.8 b A |
| Starch | 7.5 ± 0.7 a B | 8.1 ± 1.2 a A | 6.2 ± 0.0 a A | 6.2 ± 0.0 a B |
| Malic acid | 6.2 ± 0.9 b A | 7.5 ± 1.5 a A | 1.8 ± 0.0 a B | 1.5 ± 0.0 a B |
| Ascorbic acid | 6.9 ± 0.7 a B | 5.7 ± 1.0 b B | 8.8 ± 0.1 a A | 7.6 ± 0.0 b A |
| Citric acid | 3.4 ± 0.1 a A | 2.9 ± 0.3 a A | 2.9 ± 0.1 a A | 3.0 ± 0.1 a A |
| Total organic acids | 16.5 ± 1.5 a A | 16.1 ± 1.8 a A | 13.5 ± 1.2 a A | 12.1 ± 1.3 a A |
| | 2nd year | | | |
| Fructose | 24.2 ± 0.3 a B | 17.4 ± 0.5 b B | 70.9 ± 1.1 a A | 48.4 ± 0.8 b A |
| Glucose | 19.5 ± 0.1 a B | 20.5 ± 0.2 a B | 62.6 ± 0.8 a A | 45.5 ± 0.8 b A |
| Sucrose | 7.1 ± 0.1 a B | 3.5 ± 0.1 b A | 4.6 ± 0.1 a A | 3.4 ± 0.1 b A |
| Inositol | 1.2 ± 0.0 a A | 1.1 ± 0.0 a A | 0.8 ± 0.0 a A | 0.9 ± 0.0 a A |
| Total sugars | 52.0 ± 0.4 a B | 42.5 ± 0.7 b B | 138.9 ± 1.9 a A | 98.3 ± 1.7 b A |
| Sweetness index | 8.5 ± 0.2 a B | 6.6 ± 0.4 b B | 23.3 ± 2.5 a A | 16.2 ± 2.1 b A |
| Starch | 8.2 ± 0.1 a A | 8.4 ± 0.1 a A | 6.3 ± 0.2 a B | 5.9 ± 0.5 a A |
| Malic acid | 4.7 ± 0.1 b A | 5.9 ± 0.0 a A | 0.8 ± 0.1 a B | 1.4 ± 0.1 a B |
| Ascorbic acid | 6.1 ± 0.4 a B | 5.3 ± 0.2 b B | 10.4 ± 0.4 a A | 8.6 ± 0.5 a A |
| Citric acid | 3.5 ± 0.3 a A | 2.6 ± 0.1 a A | 2.6 ± 0.4 a B | 2.5 ± 0.5 a A |
| Total organic acids | 14.3 ± 3.1 a A | 13.8 ± 1.4 a A | 13.7 ± 0.6 a A | 12.5 ± 0.7 a A |

Means ± standard deviation in rows with different lowercase letters within the same measurement moment (harvest or after storage) indicate significant differences between genotypes according to Student's *t*-test at α = 0.05. Different capital letters indicate significant differences between harvest and after storage for the same genotype according to Student's *t*-test at α = 0.05.

Both genotypes exhibited a near doubling of fructose and glucose concentrations after storage, while sucrose and inositol concentrations remained at similar levels to that found at harvest (Table 6). The concentration of total sugars in "Hayward" was near 52–56 mg g$^{-1}$ FW at harvest and increased to 126–139 mg g$^{-1}$ FW after storage. Similarly, in "Arta Kiwifruit", the total sugar concentration was between 42–44 mg g$^{-1}$ FW at harvest during the two-year experimentation period and increased up to 94–98 mg g$^{-1}$ FW after storage. These findings represent an almost 147% increase in total sugars for "Hayward" kiwifruits after storage and a nearly 120% increase for "Arta Kiwifruit". The present results are within the broad range reported by Drummond [30] (11.7 g 100 g$^{-1}$ FW) and those reported by Nishiyama et al. [34], ranging from 6.15 to 10.47 g 100 g$^{-1}$ FW. Other studies have also reported total sugar values ranging from 7.7 g 100 g$^{-1}$ FW [35] to 10.8 g 100 g$^{-1}$ FW [36,37]. On a per-fruit consumption basis, one fruit of "Hayward" offers 13.31 g of sugars while one fruit of "Arta Kiwifruit" offers 11.86 g of sugars (mean of the two-year study). This is of utmost importance for a specific category of people who seek fruits of low sugar content. On the other hand, though, calculating the sweetness index (SI) of the fruit of the two genotypes based on the method of Denaxa et al. [14], "Hayward" is sweeter (SI = 20.7–23.3) than "Arta Kiwifruit" (SI = 15.6–16.2) after the storage period, which may be an advantage for people who have a preference for sweet fruits.

Kiwifruit also contains various organic acids that contribute to the fruit's overall sensory attributes, particularly in terms of the balance between sweetness and acidity. These organic acids play essential roles in both health and metabolism. Although ascorbic acid is the primary organic acid with notable health benefits in kiwifruit, other acids may

be present in high quantities too. The levels and proportions of organic acids in kiwifruit depend on the fruit's maturity, while their distribution within the fruit is not uniform [30].

As shown in Table 6, there were significant differences observed between the two genotypes in terms of organic acid concentrations, both at harvest and after storage. "Hayward" fruits exhibited significantly higher levels of ascorbic acid, ranging from 6.1 to 6.9 mg g$^{-1}$ FW (at harvest, during the two-year study) and from 8.8–10.4 mg g$^{-1}$ FW (after the storage period, during the two-year study). Park et al. [38] also noted a significant difference in ascorbic acid content among various kiwifruit cultivars. It is worth noting that the measured values of ascorbic acid in our study exceeded those reported in previous ones [24,30,34,39]. According to Esti et al. [40], factors such as genotype, ripening stage, storage conditions, and the analytical technique employed, can influence the ascorbic acid content of kiwifruit. These same authors also demonstrated that ascorbic acid content in kiwifruit samples from *Actinidia chinensis* (Planch) var. chinensis genotypes exceeded the average content found in *Actinidia deliciosa* (A. Chev) "Hayward".

Furthermore, since reductions in ascorbic acid may indicate a loss of nutritional value [41], fluctuations in ascorbic acid content during storage and ripening are critical factors influencing consumer preferences. Tavarini et al. [42] suggested that the timing of harvest significantly impacts the ascorbic acid content of the "Hayward" cultivar, with more mature fruits exhibiting lower levels. However, when these fruits were stored for 6 months and subsequently ripened, the ascorbic acid content either remained stable or increased compared to that determined at harvest. A similar pattern was observed in the current study, where both genotypes showed an approximately 46% increase in ascorbic acid concentration after storage. This finding is consistent with a study on gold kiwifruit conducted in New Zealand, which reported a substantial 17% increase in ascorbic acid content after 20 weeks of storage, with no significant changes in moisture content [30]. The per fruit consumption quantity of ascorbic acid (based on mean fruit weight determined after storage, as presented in Table 3) indicates that a single "Arta Kiwifruit" fruit offers approximately 1.03 g of ascorbic acid versus 0.96 g of a single "Hayward" fruit (based on the mean value of fruit weight and of ascorbic acid concentration determined during the two-year study).

Several studies have demonstrated that kiwifruit cultivars grown under identical geographic and climatic conditions can exhibit significant variations in their content of bioactive compounds [43,44]. Table 7 presents the results of the total phenolic compounds and antioxidant activity analysis in the studied kiwifruit genotypes. It can be observed that there were few significant differences in the measured total phenols or other phenolic compounds between the two genotypes at harvest, while their levels can be considered high [42,45]. "Hayward" fruits exhibited higher total phenol (first year) and total flavonoid (second year) content at harvest, while the opposite stood for total flavanols (second year). After storage though, "Hayward" fruits presented higher total flavonoid content (first year) while "Arta Kiwifruit" fruits had higher o-diphenol and total flavanol (second year) concentrations. "Hayward" fruits exhibited the highest level of antioxidant activity at harvest (first year), and higher antioxidant activity (ABTS assay during the second year). After storage, they exhibited the highest antioxidant activity (during the first year), but this was lower (based on the DPPH assay) during the second year. In general, fruits of both genotypes experienced a reduction of their antioxidant capacity after storage, during both years according to the literature [27,28,42]. A correlation analysis between antioxidant capacity, measured by all three assays (both at harvest and after storage), and various possible bioactive kiwifruit metabolites (total phenols, o-diphenols, flavanols, flavonoids, ascorbic acid, and carotenoids) revealed significant, positive relationships (correlation coefficient ranged between 0.32 and 0.54) in accordance with [46,47]. This indicates that any factor that could negatively influence the content of these bioactive compounds could deteriorate the nutritional value of the fruit.

**Table 7.** Effect of the genotype on kiwifruit's total phenolic compound concentration and protein content at harvest and after storage.

| Parameters | Harvest | | After Storage | |
|---|---|---|---|---|
| | Hayward | Arta Kiwifruit | Hayward | Arta Kiwifruit |
| | 1st year | | | |
| Total phenols | 0.4 ± 0.0 a A | 0.3 ± 0.0 b A | 0.4 ± 0.0 a A | 0.3 ± 0.0 a A |
| Total *o*-diphenols | 14.6 ± 0.2 a A | 12.6 ± 0.5 a A | 15.2 ± 1.4 a A | 14.7 ± 1.1 a A |
| Total flavanols | 22.4 ± 0.4 a A | 21.7 ± 0.4 a A | 15.1 ± 1.3 a B | 19.4 ± 2.7 a A |
| Total flavonoids | 6.4 ± 0.2 a A | 5.6 ± 0.1 a A | 6.2 ± 1.4 a A | 4.3 ± 1.1 b A |
| FRAP | 2.8 ± 0.5 a A | 2.3 ± 0.2 b A | 1.6 ± 0.3 a B | 1.5 ± 0.2 b B |
| DPPH | 1.9 ± 0.2 a A | 1.4 ± 0.3 b A | 1.2 ± 0.0 a B | 1.4 ± 0.2 b A |
| ABTS | 2.1 ± 0.1 a A | 1.8 ± 0.1 b A | 1.9 ± 0.5 a A | 1.3 ± 0.3 b B |
| Protein | 1.2 ± 0.0 a A | 1.4 ± 0.0 a A | 0.9 ± 0.0 a B | 1.0 ± 0.1 a A |
| | 2nd year | | | |
| Total phenols | 0.4 ± 0.0 b A | 0.4 ± 0.0 a A | 0.3 ± 0.0 a B | 0.3 ± 0.0 a B |
| Total *o*-diphenols | 16.7 ± 3.3 a A | 14.8 ± 2.2 a B | 13.0 ± 1.1 b B | 19.0 ± 1.4 a A |
| Total flavanols | 20.4 ± 1.4 b A | 22.1 ± 1.9 a A | 16.7 ± 2.7 a A | 21.4 ± 1.3 b A |
| Total flavonoids | 8.9 ± 1.6 a A | 8.5 ± 2.0 b A | 7.6 ± 1.1 a A | 5.2 ± 1.4 a B |
| FRAP | 2.9 ± 0.1 a A | 2.8 ± 0.3 a A | 2.1 ± 0.2 a B | 1.9 ± 0.3 a B |
| DPPH | 1.4 ± 0.0 b A | 1.6 ± 0.1 a A | 1.3 ± 0.1 b A | 1.9 ± 0.0 a B |
| ABTS | 2.1 ± 0.1 a A | 1.7 ± 0.1 b A | 1.9 ± 0.3 a A | 1.0 ± 0.1 b B |
| Protein | 1.1 ± 0.2 a A | 1.7 ± 0.2 a A | 0.9 ± 0.1 a A | 0.9 ± 0.1 a B |

Means ± standard deviation in rows with different lowercase letters within the same measurement moment (harvest or after storage) indicate significant differences between genotypes according to Student's *t*-test at $\alpha = 0.05$. Different capital letters indicate significant differences between harvest and after storage for the same genotype according to Student's *t*-test at $\alpha = 0.05$. Units of total phenols, mg equivalent GAE $g^{-1}$ FW; total *o*-diphenols, µg equivalent CAE $g^{-1}$ FW; total flavanols, µg equivalent CtE $g^{-1}$ FW; total flavonoids, µg equivalent CAE $g^{-1}$ FW; antioxidant capacity (FRAP, DPPH, ABTS), µmol equivalent Trolox $g^{-1}$ FW; protein, mg $g^{-1}$ FW.

Kiwifruit is not only rich in ascorbic acid, carbohydrates, and bioactive compounds but in proteins too [48]. Actinidin, a cysteine protease, makes up approximately 50% of the soluble protein content in kiwifruit [49]. The total protein concentration in kiwifruit though, is influenced by the cultivar, fruit developmental stage, and environmental conditions [48,50]. In the present study, the total protein content was found to be similar in both genotypes at harvest and after storage, as shown in Table 7. After storage though, it seemed that the protein content decreased in "Hayward" (first year) and "Arta Kiwifruit" fruits (second year), indicative of possible protease activity [48].

Kiwifruit contains significant amounts of free amino acids, which undergo significant changes as the fruit ripens [51]. In this study, we observed a decrease in some amino acids during storage, as fruit continued to mature (Table 8). Similarly, studies conducted by Drummond [30] and MacRae and Redgwell [51] indicated a reduction in most of the free amino acids during maturation, compared to immature fruits. Nonetheless, there were also some amino acids that increased after storage, such as glycine, γ-aminobutyric acid, and tryptophan (in both genotypes, during both study years). Furthermore, the major components identified in both genotypes in the present study were aspartic acid, glutamic acid, and glutamine. Drummond [30], however, found that arginine and γ-amino butyric acid (GABA) were the predominant amino acids, accounting for 36% of the total free amino acids, whereas MacRae and Redgwell [51] reported asparagine, glutamine, and arginine/GABA as the major amino acids in the whole fruit. Another study also reported the presence of tryptophan and indole amines [52], as detected in the present study.

**Table 8.** Effect of the genotype on kiwifruit's amino acids concentration ($\mu$g g$^{-1}$ FW) at harvest and after storage.

| Parameters | Harvest | | After Storage | |
|---|---|---|---|---|
| | Hayward | Arta Kiwifruit | Hayward | Arta Kiwifruit |
| | | 1st year | | |
| Alanine | 2.9 ± 0.9 b B | 4.2 ± 0.6 a A | 3.9 ± 0.9 b A | 4.9 ± 0.9 a A |
| Arginine | 7.2 ± 0.2 b A | 14.4 ± 4.5 a A | 3.9 ± 1.14 b A | 8.3 ± 1.1 a B |
| Asparagine | 15.4 ± 3.1 a A | 11.1 ± 1.2 a A | 2.2 ± 1.3 b B | 6.2 ± 0.5 a B |
| Aspartic acid | 18.5 ± 6.4 a A | 19.1 ± 4.9 a A | 19.8 ± 4.7 a A | 21.4 ± 4.3 a A |
| $\gamma$-Aminobutyric acid | 1.1 ± 0.5 a B | 1.3 ± 0.2 a B | 3.9 ± 0.6 b A | 7.6 ± 1.3 a A |
| Glutamic acid | 32.8 ± 4.2 b A | 48.2 ± 9.0 a A | 22.9 ± 7.1 b A | 38.9 ± 4.2 a B |
| Glutamine | 22.6 ± 2.4 a A | 26.1 ± 3.9 a A | 10.5 ± 3.4 b B | 14.9 ± 5.5 a B |
| Glycine | 1.1 ± 0.3 a B | 1.3 ± 0.1 a B | 6.5 ± 0.6 b A | 8.9 ± 0.6 a A |
| Isoleucine | 6.1 ± 1.4 a A | 6.5 ± 0.7 a B | 7.6 ± 1.0 a A | 7.6 ± 1.0 a A |
| Leucine | 6.7 ± 1.2 a A | 8.4 ± 1.1 a A | 4.6 ± 1.6 b A | 6.1 ± 0.6 a B |
| Methionine | 6.9 ± 0.7 a A | 7.1 ± 0.6 a A | 6.9 ± 0.3 a A | 6.8 ± 0.3 a B |
| Phenylalanine | 3.3 ± 0.3 b A | 5.7 ± 0.5 a A | 1.0 ± 0.6 b B | 1.8 ± 0.6 a B |
| Proline | 5.1 ± 0.6 a A | 4.6 ± 0.30 a A | 4.2 ± 0.5 a A | 3.1 ± 1.5 a B |
| Threonine | 2.3 ± 0.7 b A | 3.9 ± 0.8 a A | 2.9 ± 0.8 a A | 4.6 ± 1.2 a A |
| Tryptophan | 2.2 ± 2.7 b B | 3.5 ± 1.5 a B | 6.3 ± 0.4 a A | 6.5 ± 0.5 a A |
| Tyrosine | 4.9 ± 1.8 a A | 5.2 ± 1.6 a A | 3.9 ± 1.0 b A | 6.7 ± 0.3 a A |
| Valine | 6.4 ± 0.6 a A | 4.5 ± 1.0 b A | 3.2 ± 0.5 a B | 2.9 ± 1.3 a B |
| | | 2nd year | | |
| Alanine | 3.2 ± 0.9 a A | 3.8 ± 0.3 a B | 3.9 ± 0.9 a A | 5.3 ± 0.9 a A |
| Arginine | 9.4 ± 0.5 a A | 10.9 ± 1.1 a A | 7.6 ± 0.7 a A | 9.8 ± 0.7 a A |
| Asparagine | 13.1 ± 0.5 a A | 12.6 ± 2.7 a A | 2.6 ± 0.5 b B | 6.9 ± 0.9 a B |
| Aspartic acid | 25.3 ± 2.1 a A | 22.1 ± 2.6 a A | 21.4 ± 8.7 a A | 24.7 ± 7.2 a A |
| $\gamma$-Aminobutyric acid | 3.1 ± 0.1 a B | 3.8 ± 0.2 a B | 4.8 ± 0.3 b A | 9.1 ± 1.2 a A |
| Glutamic acid | 36.6 ± 6.4 b A | 44.2 ± 6.3 a A | 23.8 ± 6.2 b A | 40.8 ± 3.4 a A |
| Glutamine | 22.6 ± 0.8 a A | 23.5 ± 1.7 a A | 9.2 ± 0.7 b B | 14.5 ± 0.8 a B |
| Glycine | 1.7 ± 0.1 a B | 0.9 ± 0.2 a B | 5.6 ± 1.3 a A | 9.3 ± 2.0 a A |
| Isoleucine | 7.2 ± 0.2 a A | 6.4 ± 1.2 a B | 6.9 ± 1.0 a A | 7.7 ± 0.7 a A |
| Leucine | 6.1 ± 0.1 a A | 8.3 ± 0.0 a A | 4.5 ± 1.6 a B | 5.9 ± 1.1 a A |
| Methionine | 6.4 ± 0.1 a A | 7.9 ± 0.6 a A | 7.3 ± 0.2 a A | 7.2 ± 0.3 a A |
| Phenylalanine | 4.9 ± 0.8 a A | 4.9 ± 0.3 a A | 1.4 ± 0.5 a B | 1.8 ± 0.6 a B |
| Proline | 5.8 ± 0.7 a A | 4.2 ± 0.2 a B | 7.3 ± 1.1 a A | 7.9 ± 0.7 a A |
| Threonine | 3.6 ± 0.2 a A | 4.1 ± 0.3 a B | 2.6 ± 0.6 a A | 4.8 ± 0.8 a A |
| Tryptophan | 1.8 ± 0.8 b B | 3.4 ± 0.3 a B | 8.3 ± 0.4 a A | 8.4 ± 0.5 a A |
| Tyrosine | 4.7 ± 0.2 a A | 5.3 ± 0.2 a B | 4.2 ± 1.0 b B | 6.8 ± 1.2 a A |
| Valine | 6.8 ± 0.9 a A | 5.5 ± 1.3 b A | 4.4 ± 1.2 a A | 2.8 ± 0.8 a B |

Means ± standard deviation in rows with different lowercase letters within the same measurement moment (harvest or after storage) indicate significant differences between genotypes according to Student's *t*-test at $\alpha$ = 0.05. Different capital letters indicate significant differences between harvest and after storage for the same genotype according to Student's *t*-test at $\alpha$ = 0.05.

As shown in Table 8, alanine, arginine, glutamic acid, phenylalanine, threonine, and tryptophan were significantly higher in "Arta Kiwifruit" compared to "Hayward" at harvest of the first year, while valine concentration was higher in "Hayward" fruits during both study years. After storage, "Arta Kiwifruit" exhibited significantly higher concentrations of alanine, arginine, asparagine, $\gamma$-aminobutyric acid, glutamic acid, glutamine, glycine, leucine, phenylalanine, and tyrosine compared to "Hayward" fruits during the first year and of asparagine, $\gamma$-aminobutyric acid, glutamic acid, and glutamine, during the second year.

PCA was used to examine the differences between the genotypes based on fruit quality parameters measured at harvest and after the storage period, separately per study year.

As can be seen both during harvest and after the storage period of the first year (Figure 1A,B), the two genotypes separated as "Hayward" were located at the right side

of the principal component (PC) 1, while "Arta Kiwifruit" was located at the left side of PC1 at harvest, while the exact opposite stood after storage. At harvest, "Hayward" was characterized by a high concentration of phenolic compounds, carotenoids, sugars, and antioxidant capacity (based on DPPH and FRAP assays), while "Arta Kiwifruit" was characterized by high fruit weight, firmness, length, and starch, among others.

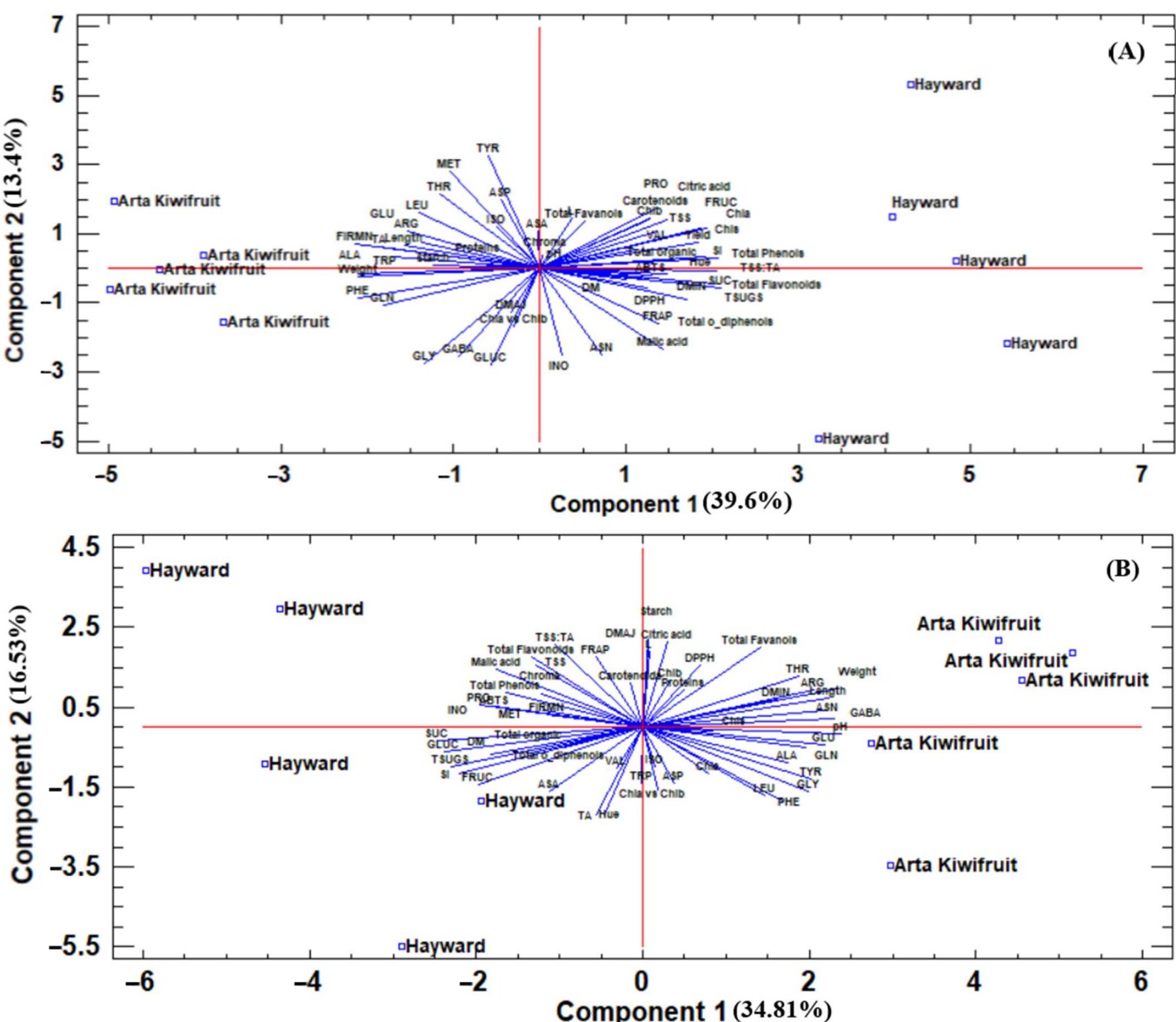

**Figure 1.** Biplot presentation of the PCA analysis of the raw data of the first year, at harvest (**A**) and after the storage period (**B**). Abbreviations: Chl, chlorophyll; TRP, tryptophan; ASP, aspartic acid; ISO, isoleucine; LEU, leucine; PHE, phenylalanine; GLY, glycine; TYR, tyrosine; ALA, alanine; GLN, glycine; GLU, glutamic acid; ASN, asparagine; GABA, γ-aminobutyric acid; DMIN, diameter minor; ARG, arginine; THR, threonine, DMAJ, diameter major; TSS, total soluble solids; PRO, proline; MET, methionine; FIRMN, firmness; SUC, sucrose; DM, dry mass; GLUC, glucose; TSUGS, total sugars; SI, sweetness index; FRUC, fructose; ASA, ascorbic acid; VAL, valine; TA, titratable acidity; DPPH, antioxidant capacity based on diphenyl picryl hydrazyl assay; FRAP, ferric reducing antioxidant power; ABTS, antioxidant capacity based on 2,2-azino-bis (3-ethylbenzothiazoline-6-sulfonic acid assay.

After storage (Figure 1B) "Hayward" was characterized by high sugar concentration, antioxidant activity (by FRAP and ABTS assays), and TSS, among others, while "Arta Kiwifruit" was characterized by high fruit weight, some amino acids, and pH.

Similarly, as before, the two genotypes were clearly separated during the second year, both at harvest (Figure 2A) and after the storage period (Figure 2B). At harvest, "Hayward" was located at the right side of PC1 exhibiting high sugar concentration, high TSS:TA ratio, and high antioxidant capacity (based on FRAP and ABTS assays). "Arta Kiwifruit", on the other hand, was located at the left side of PC1 presenting high fruit weight and dimensions, high antioxidant capacity (based on DPPH assay), and a high concentration of some amino acids such as threonine, tryptophan, glutamic acid, etc.

After the storage period, this clear separation still existed, as "Hayward" was located on the left side of PC1 and on the right the "Arta Kiwifruit" (Figure 2B). "Hayward" was characterized by high TSS and TSS:TA ratio, total sugars, and antioxidant capacity (by FRAP and ABTS assays), while "Arta Kiwifruit" was characterized by high fruit weight, total flavanol, and *o*-diphenol concentrations, and antioxidant capacity (by DPPH assay), among others.

The phenotypic description of the two genotypes revealed differences concerning shoot, leaf, and fruit morphological indexes, as shown in Table 9. "Hayward" exhibited strong coloration of the growing tip of the young shoots as well as strong coloration of the upper side of the petiole. "Arta Kiwifruit", on the other hand, presented higher fruit weight, while its fruit was characterized by an oblong shape and rounded stalk end. On the contrary, "Hayward" fruit was lighter and elliptic with a flat stylar end, but sweeter than that of "Arta Kiwifruit".

**Table 9.** The most significant differences between the two genotypes, based on UPOV key morphological traits.

| Studied Characteristics (UPOV Description) | Genotypes | | | |
|---|---|---|---|---|
| | Hayward | | Arta Kiwifruit | |
| | Characterization | Score | Characterization | Score |
| Young shoot: density of hairs | Medium | 5 | Sparse | 3 |
| Young shoot: anthocyanin coloration of growing tip | Strong | 7 | Medium | 5 |
| Petiole: anthocyanin coloration of upper side | Strong | 7 | Medium | 5 |
| Fruit: weight | High | 7 | Very high | 9 |
| Fruit: length | Medium | 5 | Long | 7 |
| Fruit: ratio length/width | Medium | 5 | Weakly elongated | 3 |
| Fruit: shape | Elliptic | 3 | Oblong | 2 |
| Fruit: stylar end | Flat | 3 | Rounded | 4 |
| Fruit: shape of shoulder at stalk end | Weakly sloping | 2 | Truncate | 1 |
| Fruit: length of stalk relative to length of fruit | Long | 7 | Medium | 5 |
| Fruit: width of core relative to fruit | Large | 7 | Very large | 9 |
| Fruit: sweetness | Low | 3 | Very low | 2 |
| Time of beginning of flowering | Late | 7 | Very late | 8 |
| Time of maturity for harvest | Late | 7 | Very late | 8 |

The extracted DNA exhibited sufficient quality and quantity for genetic analysis. The electrophoresis results of the PCR products are presented in Figure 1. The framed figures and arrows highlight the observed genetic differences between the two genotypes. The two genotypes, i.e., the "Hayward" cultivar and the Greek genotype "Arta Kiwifruit", were sufficiently distinguished by UBC-844 and UBC-810 ISSR markers. Different amplification bands were observed between the two genotypes, as indicated by the electrophoresis results (Figures 3 and 4). ISSRs have proven effective in distinguishing kiwifruit genotypes [53] and assessing the genetic stability of micropropagated Actinidia plants [13], thereby solidifying their pivotal role as a valuable molecular tool in kiwifruit breeding programs.

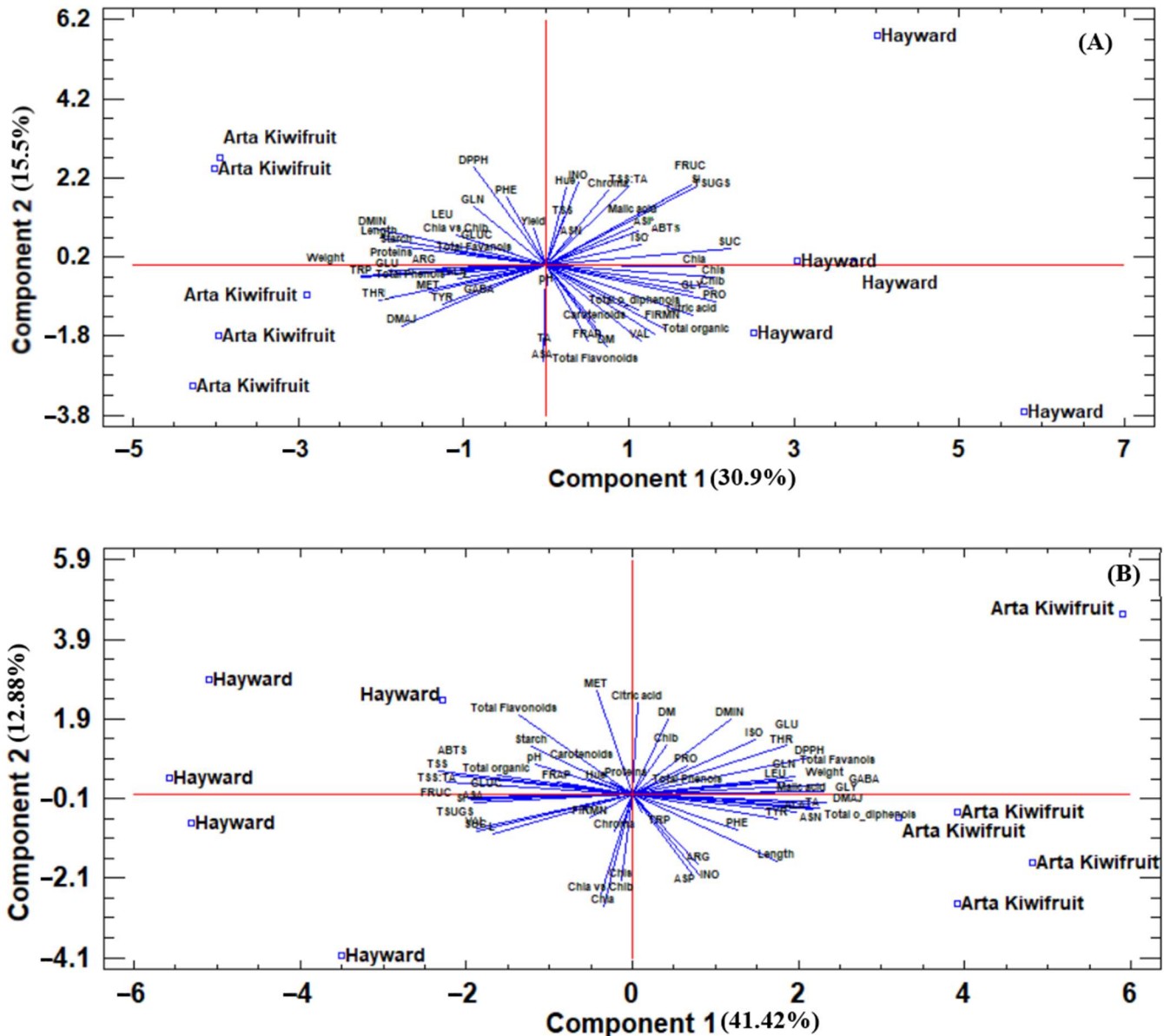

**Figure 2.** Biplot presentation of the PCA analysis of the raw data of the second year, at harvest (**A**) and after the storage period (**B**). Abbreviations: Chl, chlorophyll; TRP, tryptophan; ASP, aspartic acid; ISO, isoleucine; LEU, leucine; PHE, phenylalanine; GLY, glycine; TYR, tyrosine; ALA, alanine; GLN, glycine; GLU, glutamic acid; ASN, asparagine; GABA, γ-aminobutyric acid; DMIN, diameter minor; ARG, arginine; THR, threonine, DMAJ, diameter major; TSS, total soluble solids; PRO, proline; MET, methionine; FIRMN, firmness; SUC, sucrose; DM, dry mass; GLUC, glucose; TSUGS, total sugars; SI, sweetness index; FRUC, fructose; ASA, ascorbic acid; VAL, valine; TA, titratable acidity; DPPH, antioxidant capacity based on diphenyl picryl hydrazyl assay; FRAP, ferric reducing antioxidant power; ABTS, antioxidant capacity based on 2,2-azino-bis (3-ethylbenzothiazoline-6-sulfonic acid assay.

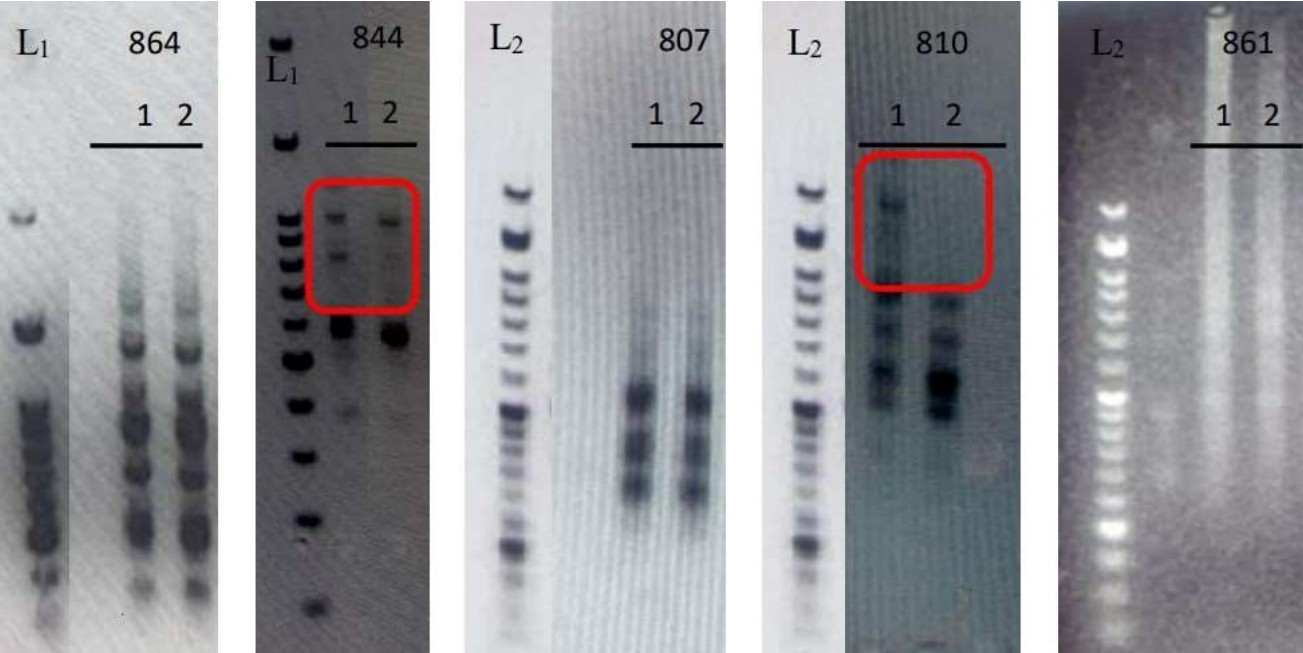

**Figure 3.** ISSR banding profile generated by UBC-864, UBC-844, UBC-807, UBC-810, and UBC-861. The red frame shows the genetic differences between the two genotypes concerning the presence of amplification bands. L$_1$: Ladder 3000 bp; L$_2$: 1500 bp; 1: "Arta Kiwifruit"; 2: "Hayward" cultivar.

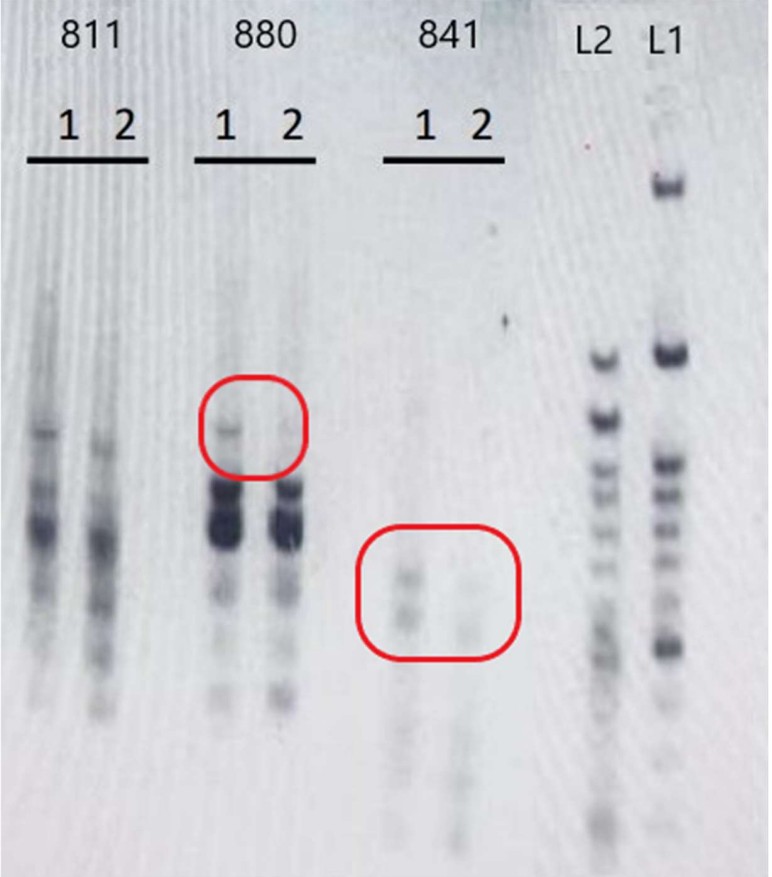

**Figure 4.** ISSR banding profile generated by UBC-811, UBC-880, and UBC-841. The red frames show the genetic differences between the two genotypes concerning the presence of amplification bands. L$_1$: Ladder 3000 bp; L$_2$: 1500 bp; 1: "Arta Kiwifruit"; 2: "Hayward" cultivar.

## 4. Conclusions

In summary, the comprehensive analysis conducted in this study, encompassing both biochemical assessments and a phenotypical description according to UPOV key morphological parameters, in conjunction with the successful implementation of ISSR molecular markers, unequivocally demonstrates that the Greek genotype "Arta Kiwifruit", exhibits a distinct genetic profile in comparison to the widely cultivated "Hayward" cultivar. Notably, substantial variations in key morphological attributes, particularly in fruit size and shape, have been observed, as "Arta Kiwifruit" produced larger fruits of good quality characteristics during both study years (based on TSS content, firmness, and dry mass at harvest). Furthermore, it exhibited satisfactory behavior during the storage period, comparable to this of the "Hayward" cultivar, while it was characterized by higher amino acid concentration compared to the "Hayward" cultivar. These distinctive characteristics hold immense promise, rendering "Arta Kiwifruit" potentially attractive to both growers and consumers, providing a competitive advantage within the market.

The results of our research endorse the commercial viability of "Arta Kiwifruit", encouraging further breeding endeavors for future commercial production. Furthermore, the identification and validation of this novel kiwifruit genotype carry significant implications, not only for the local kiwifruit industry but also on a global scale, potentially paving the way for the breeding of new cultivars that incorporate desirable traits that were identified.

**Author Contributions:** Conceptualization, P.A.R., N.-K.D., A.T. and E.N.; methodology, P.A.R.; software, N.-K.D., A.T., E.N. and A.K.; validation, P.A.R., N.-K.D. and E.T.; formal analysis, P.A.R. and N.-K.D.; investigation, P.A.R., N.-K.D., A.T., E.N., A.K., E.S. and E.T.; resources, P.A.R.; data curation, P.A.R., N.-K.D. and A.K.; writing—original draft preparation, P.A.R. and N.-K.D.; writing—review and editing, P.A.R.; visualization, N.-K.D.; supervision, P.A.R.; project administration, P.A.R.; funding acquisition, P.A.R. All authors have read and agreed to the published version of the manuscript.

**Funding:** This research was funded by the framework of the call for proposals "Support to SMEs for research projects in the fields of agrifood, health and biotechnology" under the Operational Program "Epirus 2014–2020" with the project code ΗΠ1ΑΒ-00195.

**Data Availability Statement:** The data presented in this study are available on request from the corresponding author. The data are not publicly available due to privacy restrictions.

**Acknowledgments:** We are thankful to Christoforos Xylogiannis and Evanthis Xylogianni for their great collaboration and to Christos Kollios (Kollios Fruit Trading and Export S.A.) for allowing us to use his storage facilities.

**Conflicts of Interest:** The authors declare no conflict of interest.

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
