# Peer review of "Physical, Organoleptic, and Phytochemical Valuation of the Promising Greek Kiwifruit Genotype Arta Kiwifruit"

_horticulturae, doi:10.3390/horticulturae9121276_

Round 1
Reviewer 1 Report
Comments and Suggestions for Authors
Summary
This study is focused on the evaluation of the quality and biochemical features of a new genotype of kiwifruit selected in Grece, as compared to those of the renown cv Hayward. Phenotypic traits and molecular markers of the two genotypes are also assessed and compared. The final objective is to highlight valuable characteristics of the new genotype in order to consider the usefulness of its commercial diffusion.
General Comments
Strengths
The article provides information and details on fruit quality, biochemical, phenotypic and molecular characteristics of a new promising kiwifruit cultivar.
The introduction is clear and pertinent with the topic.
Physical, chemical and molecular analysis are well described and conducted.
Weaknesses
Data collected in the two years of trial are presented as a mean value. It is therefore not possible to understand whether there is any repeatability in the values of the fruit quality parameters or whether, viceversa, they fluctuate greatly between two years. Knowing this data is important for evaluating a new genotype.
The average productivity of the plants of the two cultivars is not reported. Knowledge of this data should not be overlooked, since the production level is fundamental for the evaluation of a new genotype and influences the fruit quality.
Among the statistical analyses, PCA has been adopted, but the results are commented superficially and incompletely.
Specific comments
Many specific suggestions to improve the text of the manuscript are directly reported in the pdf. file.
Here are some comments.
Abstract
Line 16. Do you mean commercial chemical characteristics? But there was a difference in TSS concentration! Please clarify this point.
Introduction
Line 46, 74, 75, 78. The name of a plant should not have quotation marks when accompanied by the words “cultivar”, “variety” “genotype” etc., or their abbreviations.
Materials and Methods
Line 83. See comment in the Introduction section.
Line 87. This is a repetition: please eliminate it.
Line 96. In my opinion, the properties that were analyzed concern the fruit quality, not the physiology.
Line 107-109. As concerns the color, please specify here the color system and coordinates that you used (CIE L*a*b*), reference for calculation of Chroma and Hue indices, and what all these parameters represent.
Lines 120-124. In my opinion, juice TSS and TA are part of the commercial fruit quality traits.
Lines 219-220. Since only two genotypes are compared, the ANOVA is exhaustive, i.e., no other tests are needed to discriminate the differences.
224. Did you repeat the phenotypic and molecular characterization in both years? If not, you must specify when you carried out these trials.
Results and Discussion
For each parameter analyzed in the two years, the results obtained in each year must be shown, not just the two-year average values, in order to verify the repeatability of the data and the extent of the variations.
Lines 266-267. Sotiropoulos et al. compared different varieties, so his findings do not confirm your results. Moreover, the existence of differences among fruits of varieties is obvious.
Line 270. What do you mean as “consistent shape”? Please, clarify.
Line 285. Please, report literature for this limit.
Lines 300-301; 325-326; 362-363; 432-433; 455-456. Please, see note at Tab. 3.
Lines 351-352. The meaning of this phrase is unclear. It needs to be explained better.
Lines 458-459. I disagree with the statement that PCA shows no differences between cultivars. There are areas of prevalence of one or the other cultivar and areas where there is overlap. We should comment.
Lines 457-458. With PCA analysis it is necessary to specify more clearly the parameters associated with each component.
Line 474-475. The results should be (at least briefly) commented.

Comments on the Quality of English LanguageThe quality of English Language is quite fine, but can be improved.
Author Response
Dear reviewer,
Please find attracted a point-to-point response to your remarks and suggestions yellow highlighted in the new manuscript.
Best Regards,
Roussos A. Petros,
Denaxa Nikoleta-Kleio
(on authors’ behalf)

Reviewer 2 Report
Comments and Suggestions for Authors
The paper presents a comprehensive research and skills on the new kiwifruit genotype ‘Arta Kiwifruit’. To improve it, I suggest:
Simplify the language in the introduction to make it more accessible.
Highlight the practical implications of the findings in the summary.

Author Response
Dear reviewer,
Please find attracted a point-to-point response to your remarks and suggestions green highlighted in the new manuscript.
Best Regards,
Roussos A. Petros,
Denaxa Nikoleta-Kleio
(on authors’ behalf)

Reviewer 3 Report
Comments and Suggestions for Authors
The manuscript: "Evaluation of a promising Greek kiwifruit genotype ‘Arta Kiwifruit’" describes morphological, physiological and biochemical characteristics of a new kiwifruit genotype ‘Arta Kiwifruit’ compared to the ‘Hayward’ cultivar. In addition, the genetic analysis by ISSR markers confirmed distinction between two tested genotypes. This study could have practical application in the kiwifruit breeding programs for the obtainment of new cultivars with elite traits.
The following clarifications and changes are recommended and should be make:
Title
The title of the manuscript is too general and did not represent the main subject of the evaluation.
Abstract
- The aim of the study should be clearly presented. It should be mentioned in the abstract that parameters related to physiological characteristics and biochemical composition were performed at harvest and after storage.
- The ISSR as an abbreviation should be defined in the brackets.
Introduction
- The authors should emphasized in the Introduction section some key findings of previous studies (References number 12 and 13) for using ISSR markers in the identification and classification of different kiwifruit cultivars.
Materials and methods
- The methods used for determination of Total phenolics, diphenols, flavanols and flavonoids should be presented. For example, total phenolic content was analyzed by Folin-Ciocalteu method…
Results and Discussion
Table 3-Table 8: Why the authors did not present standard deviation along with the mean values of the analyzed parameters?
Pg. 6, Line 261-263: The following sentence should be revised: „Notably, the mean fruit weight of ‘Arta Kiwifruit’ was significantly higher compared to ‘Hayward’ at both harvest and after storage, weighing 138 g and 125 g, respectively.“. The values 138 g and 125 g for fruit weight correspond only for Arta Kiwifruit at harvest and after storage, respectively.
Pg. 6, Line 263-264: In the following statement, it is not mentioned is this comparison correspond to „at harvest“ or „after storage“: „This represents a substantial difference, with ‘Arta Kiwifruit’ fruits being approximately 34.62% heavier compared to ‘Hayward’.“
- The authors should clearly present the results. The comparison of the analyzed parameters should be: between two genotypes at harvest, between two genotypes after storage, as well for each genotype between harvest and after storage. This could provide better representation of physiological and biochemical differences between genotypes, as well for variations that occur during storage.
Pg. 4, Line 169-177: There is a mistake in the following indication for Table. It should be Table 4: „Furthermore, both ‘Hayward’ and ‘Arta Kiwifruit’ are green-fleshed kiwifruits, sharing a similar hue angle and they do not exhibit significant differences in terms of their flesh color parameters (Table 5).“
Pg. 10, Line 416-418: The following statement is not clear, it should be revised: „It can be observed that there were no significant differences in the measured total phenols or other phenolic compounds between the two genotypes, while their levels can be considered high [41,44].“
Pg. 10, Line 413-421: The authors should discuss the relationship between analyzed phenolic compounds and antioxidant assays or potential contribution of phenolics to the antioxidant activities in both genotypes. What about other studies for antioxidant activities of kiwifruit genotypes?
Table 7: The measurement units for the analyzed parameters should be provided in the Table 7, not in Table 7 Caption.
Figure 1 and Figure 2: Why the authors did not provide Loading plot of PCA with distribution of analyzed parameters on PC1 and PC2 for better visualization of the relationship between biochemical and physiological attributes, as well for characterization of the kiwifruit genotypes?
Figure 4 Caption: The Figure 4 Caption should be revised, since on the figure there is no red frame.
Pg. 13, Line 479-485: The authors should compare their results for ISSR markers with previous published data for kiwifruit.
Conclusion
The first paragraph of the Conclusion section needs revision and it should represent the key findings of the study.
Comments on the Quality of English LanguageMinor editing of English language is necessary.
Author Response
Dear reviewer,
Please find attracted a point-to-point response to your remarks and suggestions cyan highlighted in the new manuscript.
Best Regards,
Roussos A. Petros,
Denaxa Nikoleta-Kleio
(on authors’ behalf)

Round 2
Reviewer 3 Report
Comments and Suggestions for Authors
The authors appropriately responded to all suggestions.